# Assessment of Spinal Muscular Atrophy Carrier Status by Determining *SMN1* Copy Number Using Dried Blood Spots

**DOI:** 10.3390/ijns6020043

**Published:** 2020-05-29

**Authors:** Yogik Onky Silvana Wijaya, Jamiyan Purevsuren, Nur Imma Fatimah Harahap, Emma Tabe Eko Niba, Yoshihiro Bouike, Dian Kesumapramudya Nurputra, Mawaddah Ar Rochmah, Cempaka Thursina, Sunartini Hapsara, Seiji Yamaguchi, Hisahide Nishio, Masakazu Shinohara

**Affiliations:** 1Department of Community Medicine and Social Healthcare Science, Division of Epidemiology, Kobe University Graduate School of Medicine, 7-5-1 Kusunoki-cho, Chuo-ku, Kobe 650-0017, Japan; yogik.onky@gmail.com (Y.O.S.W.); niba@med.kobe-u.ac.jp (E.T.E.N.); mashino@med.kobe-u.ac.jp (M.S.); 2Medical Genetics Laboratory, National Center for Maternal and Child Health, Khuvisgalchdyn Street, Bayangol District, Ulaanbaatar 16060, Mongolia; p_jamiyand@yahoo.com; 3Department of Clinical Pathology and Laboratory Medicine, Faculty of Medicine, Universitas Gadjah Mada, Radiopoetro Building 5th floor, Jl. Farmako, Sekip Utara, Yogyakarta 55281, Indonesia; imma.harahap@ugm.ac.id; 4Faculty of Nutrition, Kobe Gakuin University, 518 Arise, Ikawadani-cho, Nishi-ku, Kobe 651-2180, Japan; bouike@nutr.kobegakuin.ac.jp; 5Department of Pediatrics, Faculty of Medicine, Universitas Gadjah Mada, Jl. Kesehatan No.1, Sekip, Yogyakarta 55281, Indonesia; dian.k.nurputra@ugm.ac.id (D.K.N.); sunartini_hapsara@ugm.ac.id (S.H.); 6Department of Neurology, Faculty of Medicine, Universitas Gadjah Mada, Jl. Kesehatan No.1, Sekip, Yogyakarta 55281, Indonesia; mawaddah.arr@gmail.com (M.A.R.); cempakathursina@ugm.ac.id (C.T.); 7Department of Pediatrics, Shimane University School of Medicine, 89-1 Enya, Izumo, Shimane 693-8501, Japan; seijiyam@med.shimane-u.ac.jp; 8Faculty of Rehabilitation, Kobe Gakuin University, 518 Arise, Ikawadani-cho, Nishi-ku, Kobe 651-2180, Japan

**Keywords:** spinal muscular atrophy, carrier, *SMN1*, dried blood spot, quantitative nested PCR

## Abstract

Spinal muscular atrophy (SMA) is a common neuromuscular disease with autosomal recessive inheritance. The disease gene, *SMN1*, is homozygously deleted in 95% of SMA patients. Although SMA has been an incurable disease, treatment in infancy with newly developed drugs has dramatically improved the disease severity. Thus, there is a strong rationale for newborn and carrier screening for SMA, although implementing SMA carrier screening in the general population is controversial. We previously developed a simple, accurate newborn SMA screening system to detect homozygous *SMN1* deletions using dried blood spots (DBS) on filter paper. Here, we modified our previous system to detect the heterozygous deletions of *SMN1*, which indicates SMA carrier status. The system involves a calibrator-normalized relative quantification method using quantitative nested PCR technology. Our system clearly separated the DBS samples with one *SMN1* copy (carrier status with a heterozygous deletion of *SMN1*) from the DBS samples with two *SMN1* copies (non-carrier status with no deletion of *SMN1*). We also analyzed DBS samples from SMA families, confirmed SMA in the affected children, and determined the carrier status of their parents based on the *SMN1* copy number. In conclusion, our system will provide essential information for risk assessment and genetic counseling, at least for SMA families.

## 1. Introduction

Spinal muscular atrophy (SMA) is an autosomal recessive and progressive neuromuscular disease characterized by muscle weakness and atrophy that results from the degeneration of motor neurons in the spinal cord [1]. The incidence of SMA is 1 in 6000 to 10,000 live births [1], and the carrier frequency is 1 in 40 to 60 in the general population [1]. Survival Motor Neuron 1 (*SMN1*) has been identified as the SMA-causing gene [2]. *SMN1* is absent (or homozygously deleted) in ~95% of SMA patients and deleteriously mutated in some of the remaining patients [2,3]. Thus, a deletion test for *SMN1* is the first-tier evaluation for SMA diagnosis. 

SMA phenotypes are classified by onset age and achieved motor milestones. Patients with SMA type 1 develop symptoms in the first 6 months after birth, never achieve the motor milestone of sitting independently, and have a life expectancy of less than 2 years without respiratory support [4]. SMA type 1 constitutes the largest group of SMA patients, and it is the most frequent genetic cause of death in infants.

SMA is an incurable disease. However, Spinraza^®^ (nusinersen, an antisense oligonucleotide-based SMN protein-enhancing therapy) and Zolgensma^®^ (onasemnogene abeparvovec-xioi, adeno-associated virus vector-based *SMN1*-gene therapy) have been approved by regulatory agencies in multiple countries, including the U.S. and Japan, as treatment choices for SMA. According to their clinical trials, these drugs greatly improved the disease severity; more specifically, they facilitated motor milestone achievement and reduced the number of cases requiring respiratory support [5,6]. 

It should be noted that these drugs achieved a better clinical outcome for SMA patients when initiated in early infancy [5,6,7]. For an early diagnosis and the initiation of treatment, newborn screening or carrier screening may be useful, and they are feasible because the majority of the patients lack *SMN1* [8,9]. SMA infants can be detected by having no copies of *SMN1* and SMA carriers by having one copy. 

Several pilot studies have been reported that describe newborn screening for SMA using dried blood spots (DBS) on filter paper [9,10,11,12,13]. Vill et al. stated, based on their experience of SMA newborn screening and presymptomatic treatment, that newborn screening improved the outcome for children with genetically proven SMA, and concluded that newborn screening for SMA should be introduced in all countries where therapy is available [14].

SMA carrier screening has been widely discussed. The American College of Medical Genetics issued a statement saying that “carrier testing should be offered to all couples” [8,9]. However, population carrier screening for SMA may not be cost-effective [15] and future parents may be unprepared to accept screening results [16]. For families with an SMA patient, especially for couples who desire to bear children in such families, carrier diagnosis is highly useful because carrier screening can inform genetic counseling and reproductive options for high-risk couples.

Early treatment in infancy with newly developed drugs has dramatically improved the disease severity; therefore, there is a strong rationale for newborn screening and carrier screening for SMA. In this study, we developed a new simple, accurate, and inexpensive system for determining the *SMN1* copy number and SMA carrier status. This system is, in a sense, a modified version of our system to detect the homozygous deletion of *SMN1* [13]. Our current system, which uses a dried blood spot (DBS) on filter paper, may provide critical information to SMA families all over the world.

## 2. Material and Methods 

### 2.1. Objectives and Ethics

We have established a simple system using DBS for SMA newborn screening [13]. In this study, we aimed to establish another simple system using DBS to screen for SMA carrier status. Our primary objective was to develop a system to detect heterozygous *SMN1* deletion using DBS from the parents of *SMN1*-deleted SMA patients. The second objective was to confirm the practicality of our system to detect homozygous and heterozygous *SMN1* deletion using DBS on filter paper in resource-limited scenarios where molecular genetic analysis is not available. This study was approved by the institutional review boards at all participating hospitals, as well as the Ethics Committee of the Kobe University Graduate School of Medicine (reference 1089, approved on 5 October 2018), and was conducted in accordance with the World Medical Association Declaration of Helsinki.

#### 2.1.1. DBS Samples for Validation of *SMN1* Copy Number Analysis

As media for DBS, we used FTA (Flinders Technology Associates) Elute Cards^®^ (GE Healthcare, Boston, MA, USA) and Guthrie cards (Toyo Roshi No. 545 filter paper, Toyo Roshi Kaisha, Tokyo, Japan). The FTA Elute Cards^®^ were developed for long-term DNA preservation at room temperature, and Toyo Roshi No. 545 filter paper was used in neonatal screening.

For the validation of the system, we analyzed 98 FTA cards with DBS (44 SMA carriers with one *SMN1* copy and 54 controls with two *SMN1* copies) and 33 Guthrie cards with DBS (12 SMA carriers with one *SMN1* copy and 21 controls with two *SMN1* copies) from the sample library of the Division of Epidemiology, Kobe University Graduate School of Medicine. 

The SMA carriers in the validation study had been shown to carry one copy of *SMN1* using the method of Tran et al. and fresh blood [17]. Similarly, the validation study controls had been determined to carry two copies of *SMN1* using fresh blood. 

#### 2.1.2. DBS Samples from Mongolian and Indonesian SMA Families

To detect *SMN1* deletion, samples from Mongolian and Indonesian SMA patients and their parents were analyzed using the method of Shinohara et al. [13]. Patient 1 was a 21-month-old boy who showed delayed motor milestones and muscle weakness. He was clinically diagnosed as having SMA type 1. Three Guthrie cards with DBS from Patient 1 and his parents were used in this study. Patients 2 and 3 were 7-month-old twin boys who showed muscle weakness at three months old. Their brother, who died at two months old, had already been diagnosed as having SMA by a molecular genetic analysis. These siblings were clinically diagnosed as having SMA type 1. Two Guthrie cards with DBS from Patients 2 and 3 were used in this study. Patient 4 was a 2-year-old girl, who showed muscle weakness and delayed motor milestones. She was clinically diagnosed as having SMA type 2. Three Guthrie cards with DBS from Patient 4 and her parents were used in this study.

### 2.2. Detection of Homozygous SMN1 Deletion in DBS Samples

We performed *SMN1* deletion tests with DBS on FTA cards and Guthrie cards using the previously described procedures [13]. Our procedure outline was: (1) a punched DBS circle was placed directly into conventional PCR reaction mixture, followed by the pre-amplification of the *SMN1/SMN2* exon 7-targeted region with 40 cycles (to reach the plateau phase in the first round PCR). (2) The pre-amplification products were diluted 100-fold. (3) *SMN1* exon 7 was specifically amplified from the diluted pre-amplified product by 20 cycles of real-time modified competitive oligonucleotide priming-PCR (real-time mCOP-PCR) [18] (to detect the presence or absence of *SMN1* in the second round PCR).

### 2.3. Detection of Heterozygous SMN1 Deletion in DBS Samples

#### 2.3.1. Outline

We performed an *SMN1* copy number analysis using DBS on FTA cards and Guthrie cards. To determine the *SMN1* copy number, we adopted a “calibrator-normalized relative quantification method”. A sample from a control with two *SMN1* copies was used as a calibrator. 

Our system involves quantitative nested PCR technology. The procedure for the *SMN1* copy number analysis was: (1) a punched DBS circle from an FTA Card or Guthrie Card was placed directly into conventional multiplex PCR mixture, followed by the amplification of the targeted region of the *SMN1/SMN2* exon 7 and *CFTR* exon 4 with 20–23 cycles (to stay within the log-linear phase in the first round conventional PCR). (2) The amplification products of the first round PCR were diluted 100-fold. (3) *SMN1/SMN2* exon 7 and *CFTR* exon 4 were specifically amplified from the diluted product of the first round PCR by real-time mCOP-PCR to calculate the *SMN1* copy number based on the quantitation cycle values (Cq values) in the second round of PCR. 

#### 2.3.2. First Round PCR 

Multiplex PCR of the *SMN1/SMN2* exon 7 (target gene) and *CFTR* exon 4 (reference gene) was performed using the GeneAmp^®^ PCR System 9700 (Applied Biosystems, Foster City, CA, USA). A punched DBS circle of 2 mm in diameter was placed directly into the reaction mixture of 50 µL containing DNA polymerase KOD FX Neo (TOYOBO, Osaka, Japan). The primers for *SMN1*/*SMN2* exon 7, R111 (5′-AGA CTA TCA ACT TAA TTT CTG ATC A-3′), and 541C770 (5′-TAA GGA ATG TGA GCA CCT TCC TTC-3′), were used to amplify the target sequences of *SMN1*/*SMN2* exon 7 (Figure 1a) [2], and the primers for *CFTR* exon 4, CF621F (5′-AGT CAC CAA AGC AGT ACA GC-3′), and CF621R (5′-GGG CCT GTG CAA GGA AGT GTTA-3′), were used to amplify the target sequence of *CFTR* exon 4 (Figure 1a) [19]. The PCR conditions for a reaction mixture of 50 µL were: (1) initial denaturation at 94 °C for 7 min; (2) 20–23 cycles of denaturation at 94 °C for 1 min, annealing at 62 °C for 1 min, and extension at 72 °C for 1 min; (3) additional extension at 72 °C for 7 min; and (4) hold at 10 °C.

#### 2.3.3. Second Round PCR 

For quantitative analysis, a real-time mCOP-PCR of the *SMN1* exon 7 (target gene) and *CFTR* exon 4 (reference gene) was performed in separate tubes using the LightCycler^®^ 96 system (Roche Applied Science, Mannheim, Germany). Five microliters of 100-fold diluted first round DBS sample PCR product or 5 µL of a serially diluted sample of a calibrator sample was added up to the reaction mixture of 50 µL with DNA polymerase KOD FX Neo (TOYOBO) and EvaGreen^®^ Dye (Biotium, Hayward, CA, USA). The primers for the *SMN1*-specific amplification were R111 [2] and SMN1-COP (5′-TGT CTG AAA CC-3′) [18] (Figure 1b), and the primers for the *CFTR*-specific amplification were CF621F [19] and CFTR-COP (5′-GAG CAG TGT CCT-3′) (Figure 1b). The PCR conditions for a reaction mixture of 50 µL were: (1) initial denaturation at 94 °C for 7 min; (2) 40 cycles of denaturation at 94 °C for 30 s, annealing at 37 °C for 30 s, and extension at 72 °C for 30 s; and (3) melting curve analysis. Fluorescence signals were detected at the end of each extension procedure.

#### 2.3.4. Calculation of *SMN1* Copy Number

The amount of the first round PCR product reflected the initial number of target gene molecules (*SMN1* molecules) and reference gene molecules (*CFTR* molecules) because the amplification of the first round PCR ended in the log-linear phase. Thus, the initial number of target gene molecules (*SMN1* molecules) or reference gene molecules (*CFTR* molecules) in the second round PCR reflected the initial number of target gene molecules (*SMN1* molecules) or reference gene molecules (*CFTR* molecules) before the first round PCR. 

The *SMN1* copy number of the sample is calculated through the following formula. The formula is valid, if {ET^CqT(Sample)/ER^CqR(Sample)} is nearly equal to {ET^CqT(Calibrator)/ER^CqR(Calibrator).

Calculated value of SMN1 copy number=2×NTo(Sample)/NRo(Sample) NTo(Calibrator)/NRo(Calibrator)=2× {NTo (Sample)/NRo(Sample) }×{ET^CqT(Sample)/ER^CqR(Sample)}{NTo (Calibrator)/NRo(Calibrator) }×{ET^CqT(Calibrator)/ER^CqR(Calibrator)}=2× {NTo (Sample)× ET^CqT(Sample)}/{NRo(Sample) × ER^CqR(Sample)}{NTo (Calibrator)× ET^CqT(Calibrator)}/{NRo(Calibrator) × ER^CqR(Calibrator)}=2× NT (Sample)/NR (Sample) NT (Calibrator)/NR (Calibrator) 

Abbreviations are shown below;

NT or NR: number of target gene molecules (*SMN1* molecules) or reference gene molecules (*CFTR* molecules) at the detection threshold in the second round PCR. NTo or NRo: initial number of target gene molecules (*SMN1* molecules) or reference gene molecules (*CFTR* molecules) in the second round PCR. CqT or CqR: cycle number at the target gene (or reference gene) detection threshold in the second round PCR. ET or ER: amplification efficiency of the target gene (or reference gene) in the second round PCR. 

### 2.4. Statistical Analysis 

The 95% confidence intervals of the *SMN1* copy number values were computed using Microsoft Excel with add-in software, Statcel 3 (The Publisher OMS Ltd., Tokyo, Japan). In this report, the median, 25th percentile (first quartile), and 75th percentile (third quartile) values as well as the mean values were used to show the overlapping status of the *SMN1* copy number values between carriers and controls using Microsoft Excel.

## 3. Results

### 3.1. Detection of Homozygous SMN1 Deletion in the DBS Samples on Guthrie Cards

We have reported an *SMN1* deletion test using a combination method of targeted pre-amplification (first round PCR) and mCOP amplification (second round PCR) to detect *SMN1* [13,18]. The starting material in the previous study was DBS on FTA cards. Here, we obtained *SMN1* deletion test results using Guthrie cards instead of FTA cards.

The *SMN1* amplification curves of Patients 1, 2, 3, and 4 showed no rise before 16 cycles in the second round PCR, while those of the controls rose at a steep rate at ~10 cycles. However, the *SMN2* amplification curves of Patients 1, 2, 3, and 4 showed steep rises at ~10 cycles in the second round PCR, similar to the controls (Appendix A). These results indicated that these patients carried homozygous *SMN1* deletions, confirming the diagnosis of SMA.

The *SMN1* and *SMN2* amplification curves of the parents of Patient 1 and 4 showed steep rises at ~10 cycles in the second round PCR, which was the same as the controls. We show the amplification curves and melting peak analysis of Patient 1’s father, which are representative of all parents (Appendix A). These data showed that the father and mother were not affected by SMA, but the data did not indicate whether they were SMA carriers or not. To determine this, it is necessary to determine the *SMN1* copy number.

### 3.2. Detection of Heterozygous SMN1 Deletion in the DBS Samples

We used a quantitative nested real-time PCR assay for a copy number analysis of *SMN1*. If the amplification of the first round PCR ended in the log-linear phase, the amount of first round PCR product reflected the initial number of target gene molecules (*SMN1* molecules) or reference gene molecules (*CFTR* molecules). Similarly, if the amplification of the second round PCR ended in the log-linear phase, the amount of second round PCR product also reflected the initial number of target gene molecules (*SMN1* molecules) or reference gene molecules (*CFTR* molecules). Thus, theoretically, we are able to determine the copy number of *SMN1*.

#### 3.2.1. Amplification Efficiency of *SMN1* and *CFTR* Genes

For the copy number analysis of *SMN1*, we applied a calibrator-normalized relative quantification assay. Copy numbers were calculated using the target/reference ratio of each sample normalized by the target/reference ratio of the calibrator. The accuracy was dependent on the similar amplification efficiencies of the target and reference genes. 

In the first round PCR, we used real time PCR with serial dilution samples of genomic DNA (the starting DNA concentration was 10 ng/μL) to confirm that the amplification efficiencies of the target genes (*SMN1*/*SMN2*) and the reference gene (*CFTR*) were similar (Figure 2a) and to show that the first round PCR products reflected the ratios of the initial gene doses of *SMN1*/*SMN2* and *CFTR*. 

We also confirmed the amplification efficiencies of the second round PCR using real time PCR with serial dilution samples of the first round PCR product. We also confirmed similar amplification efficiencies for the target (*SMN1*) and reference (*CFTR*) genes (Figure 2b) to show that the second round PCR products also reflected the ratios of the initial *SMN1* and *CFTR* gene doses.

#### 3.2.2. Calculated *SMN1* Copy Number Values Using DBS on FTA Cards

Fifty-four DBS samples from controls with two *SMN1* copies and 44 DBS samples from SMA carriers with one *SMN1* copy were analyzed by a nested real-time PCR assay for a calibrator-normalized relative quantification. The results are shown with a box-and-whisker plot (Figure 3).

The ranges of the calculated *SMN1* copy number values for the 54 controls with two *SMN1* copies and the 44 SMA carriers with one *SMN1* copy were 2.12 ± 0.28 (mean ± SD) and 1.20 ± 0.15, respectively. The 95% confidence intervals were [2.11–2.20] and [1.13–1.20], respectively. In the present study with FTA cards, if the calculated *SMN1* copy number value was less than 1.40, then the *SMN1* copy number was determined as “one”.

#### 3.2.3. Calculated *SMN1* Copy Number Values Using DBS on Guthrie Cards

Twenty-one DBS samples from controls with two *SMN1* copies and 12 DBS samples from SMA carriers with one *SMN1* copy were analyzed by the same method described above, and the results are shown with a box-and-whisker plot (Figure 4).

The ranges of the calculated *SMN1* copy number values of the 21 controls with two *SMN1* copies and the 12 SMA carriers with one *SMN1* copy were 2.00 ± 0.20 (mean ± SD) and 1.15 ± 0.25, respectively. The 95% confidence intervals were [1.91–2.09] and [0.99–1.31], respectively. In the present study with Guthrie cards, if the calculated *SMN1* copy number value was less than 1.40, then the *SMN1* copy number was determined as “one”.

### 3.3. Determination of SMA Carrier Status from Analysis of SMN1 Copy Number

We determined the *SMN1* copy number in the parents of Patient 1 (Mongolian) and Patient 4 (Indonesian) using DBS samples on Guthrie cards. The calculated *SMN1* copy numbers for the father and mother of Patient 1 were 1.16 and 1.07, respectively (Figure 4). Thus, the parents were shown to be SMA carriers by molecular analysis as well as by clinical information.

The calculated *SMN1* copy number for the father of Patient 4 was 1.25 (Figure 4), indicating that he was an SMA carrier with one copy of *SMN1*. However, the mother’s calculated *SMN1* copy number was 2.09. We confirmed the presence of two *SMN1* copies in this mother with her fresh blood.

## 4. Discussion

### 4.1. Simple Method: Using Filter Paper Commonly Used for Newborn Screening

In this study, we showed that DBS on Guthrie cards can be used for the detection of both the homozygous and heterozygous deletion of *SMN1*. We had already shown that FTA cards, which are designed for the collection and storage of DNA in blood, are useful for the analysis of *SMN* genes, but we had not shown the applicability of Guthrie cards for the DNA analysis and diagnosis of genetic disorders. 

Guthrie cards (we used Toyo Roshi No. 545) are well-known for blood collection and are widely used for newborn screening for inborn errors of metabolism, including thyrotropin, phenylalanine, and 17a-hydroxyprogesterone [20], and are much cheaper than FTA cards. FTA cards are able to safely store DNA for more than 10 years, but it is not known how long Guthrie cards can be reliably kept for. The oldest DBS samples on Guthrie cards in our study were 3 years old, but we were able to determine the presence or absence of *SMN1* and the copy number of *SMN1*. DBS on Guthrie cards can be reliable genetic test samples that can be sent from anywhere to anywhere on earth at normal ambient temperatures to enable genetic testing.

However, the most distinctive feature of our system is the simple handling of DBS from FTA or Guthrie cards. All that is required is for a circle to be punched out and placed in a PCR tube; no DNA extraction procedure is necessary. In addition, the quantity and quality of DNA from DBS does not matter, which we discuss further in the next subsection.

### 4.2. Robust Method: Adopting Nested Quantitative PCR Technology

The quantity and quality of DNA from DBS varied from card to card. However, our system for the detection of the homozygous or heterozygous deletion of *SMN1* can accept low-quantity and poor-quality samples because it is based on nested PCR and a calibrator-normalized relative quantification assay with multiplex PCR.

In our previous investigation of homozygous *SMN1* deletion, we showed an example of low-quantity and poor-quality DNA from DBS samples [13]. Targeted pre-amplification by the first round PCR overcame the DNA quantity and quality problem. In our system, it was not necessary for DNA to be abundant or purified to determine the presence or absence of *SMN1*.

We also determined the gene copy number in DBS samples using nested quantitative PCR. Nested quantitative PCR, in which the product of conventional first round PCR is used as the template for a second round of quantitative real-time amplification, had been successfully used to increase assay sensitivity using DBS for the diagnosis of microorganism infection [21,22]. Nested quantitative PCR is a simple but highly sensitive technology for quantification. We applied this method to DNA from DBS to analyze the gene dosage to determine SMA carrier status.

### 4.3. Accurate Method: Adopting Calibrator-Normalized Relative Quantification Assay with Multiplex PCR 

We applied a calibrator-normalized relative quantification assay to determine the *SMN1* copy number. In a calibrator-normalized relative quantification assay, it is not necessary to know the precise copy number of the target and reference genes in the calibrator. The calibrator used in this study was a sample from a control individual with two *SMN1* copies and two *CFTR* copies. Here, only the serial dilution products of the standards from a calibrator need to be prepared; one dilution set for the target gene (*SMN1*) and one for the reference gene (*CFTR*) is required. 

We adopted a multiplex PCR in the first round PCR because the same DNA should be the template for the PCR amplification of the target and reference genes. When amplifying the target gene and the reference gene in separate tubes, the DNA should be extracted in a uniform way, accurately quantitated, and precisely added to tubes to get reliable quantification results [23]. However, such procedures are almost impossible. In addition, in our method with no DNA extraction procedures, the multiplex PCR methodology in the first round PCR was absolutely needed.

In our current method, we amplified *SMN1/SMN2* and *CFTR* in the first round PCR using the primer sets designed by Lefebvre et al. and McAndrew et al. [2,19]. We have already known the highly efficient and highly accurate amplification of *SMN1/SMN2* and *CFTR* using these “canonical” primer sets. Actually, we applied the methods of McAndrew et al. for the determination of the *SMN2* copy numbers with some modifications—for example, radioisotope-labeled primers were replaced by fluorescence-labeled primers in 2002 [24]. In those days, a large amount of pure DNA extracted from fresh blood was required for the copy number analysis of the *SMN* genes, and radioisotope-labeled or fluorescence-labeled primers were essential. In our updated methodology, a small amount of crude DNA trapped in the filter paper can be used for a gene dosage analysis, and neither radioisotope-labeled primers nor fluorescence-labeled primers are required anymore. The method described here may be in no way inferior to any methods reported previously.

### 4.4. Inexpensive Method: No Requirement for DNA Extraction or Fluorescence-Labeled Probes

Many methods for *SMN1*-specific detection have been reported [23,25,26,27,28,29,30] that can be broadly divided into two methodological groups, A and B. Methods in group A are based on the co-amplification of *SMN1* and *SMN2* with common primers, and the presence of *SMN1* is determined by an *SMN1*-specific binding of a fluorescence-labeled oligonucleotide probe [23,25,26,27,28,29]. Methods in group B are based on amplification with *SMN1*-specific primers and the detection of the amplified *SMN1* by a fluorescence-labeled oligonucleotide probe that commonly binds to *SMN1/SMN2* [30]. However, the fluorescence-labeled oligonucleotide probes in these methodologies may be expensive and their design may not be trivial. 

Our methods in previous studies [13,18] and in this study do not belong to groups A or B but to a third group. To determine the presence or absence of *SMN1*, we amplified the gene with *SMN1*-specific primers and detected its amplification by the DNA intercalation of EvaGreen^®^ Dye, which is similar to SYBR Green^®^ Dye. Our methods do not use fluorescence-labeled oligonucleotide probes, which reduces the cost. The relative cost of fluorescence-labeled oligonucleotide probes vs. the cost of EvaGreen^®^ Dye was 10:1, according to our calculation. 

The results in this study are of equal or greater accuracy compared with the methods using fluorescence-labeled oligonucleotide probes. In one study using a fluorescence-labeled probe, there was a large overlap in the ratio between normal individuals (with two *SMN1* copies) and heterozygous individuals (with one *SMN1* copy) [31]. In our study, there was no overlap in the ratio between the controls with two *SMN1* copies and SMA carriers with one *SMN1* copy (Figure 3 and Figure 4).

### 4.5. Limitation of Assigning SMA Carrier Status Based on the SMN1 Copy Number Assay

Although we are confident that our system determines accurate *SMN1* copy numbers, we also experienced a limitation encountered by other studies [9], which is the presence of SMA carriers with two *SMN1* copies. SMA is an autosomal recessive neuromuscular disorder, and ~95% of SMA patients have a homozygous deletion of the SMA-causative gene, *SMN1*. Thus, SMA carriers are usually diagnosed based on their *SMN1* copy number, because one *SMN1* copy means the heterozygous deletion of *SMN1* ([1 + 0] genotype). However, having two *SMN1* copies does not always exclude carrier status [32,33]. Some SMA carriers have two *SMN1* copies on one chromosome, with the deletion of *SMN1* on the other chromosome ([2 + 0] genotype) [32,33]. Other carriers may be heterozygous for an intragenic *SMN1* mutation ([1 + 1d] genotype; “d” denotes the presence of an intragenic mutation) [32,33].

In this study, the parents of Patient 1 and the father of Patient 4 showed one *SMN1* copy, indicating that they were SMA carriers with a [1 + 0] genotype. However, the mother of Patient 4 showed two *SMN1* copies. Patient 4 was homozygous for *SMN1* deletion, which means that she inherited a chromosome with no *SMN1* alleles from her mother. Therefore, Patient 4’s mother had two *SMN1* copies on one chromosome and no *SMN1* copy on the other chromosome, which was inherited by her daughter. These results indicated a limitation of SMA carrier diagnosis based on the gene copy number analysis of *SMN1*. 

The current screening method based on an *SMN1* copy number analysis could not detect SMA carriers with a [2 + 0] genotype or [1 + 1d] genotype, although their frequencies are low. The occurrences of [2 + 0] and [1 + 1d] genotypes have been presumed to be of low incidence among SMA carriers. According to a previous report, ~2.4% of carriers showed the [2 + 0] genotype, and ~1.7% of carriers showed the [1 + 1d] genotype [34].

Even so, how to solve the issues regarding SMA carriers with a [2 + 0] or [1 + 1d] genotype should be explored. Some researchers observed that some variants were related to duplications of *SMN1* in SMA families [35,36]. The identification of such variants and their haplotype analysis may be helpful for the risk estimation of SMA carriers with [2 + 0] genotype. As for the screening of SMA carriers with [1 + 1d], next generation sequencing has the potential to detect *SMN1* point mutations in the newborn or carrier screening programs [37]. 

## 5. Conclusions

We developed a new simple, accurate, and inexpensive system for determining the *SMN1* copy number and SMA carrier status. The system developed in this study has three main advantages. First, Guthrie cards as well as FTA cards can be used as simple DNA collection and storage media. Guthrie cards are well-known filter papers that are used for screening congenital metabolic disorders. With Guthrie cards, DNA can be delivered from any remote area where genetic analysis is not available to a DNA-analyzing center. Second, the procedures in our system are simple but accurate for determining the *SMN1* copy number. This provides critical information for the genetic counselling of SMA families. Finally, our system does not require DNA extraction kits or oligonucleotide probes. Our system is significantly cheaper than other methods, which becomes more important when analyzing large numbers of samples. These characteristics of our system may allow a greater number of SMA families all over the world to receive the correct assessment of the recurrence risk of the disease.

## Figures and Tables

**Figure 1 IJNS-06-00043-f001:**
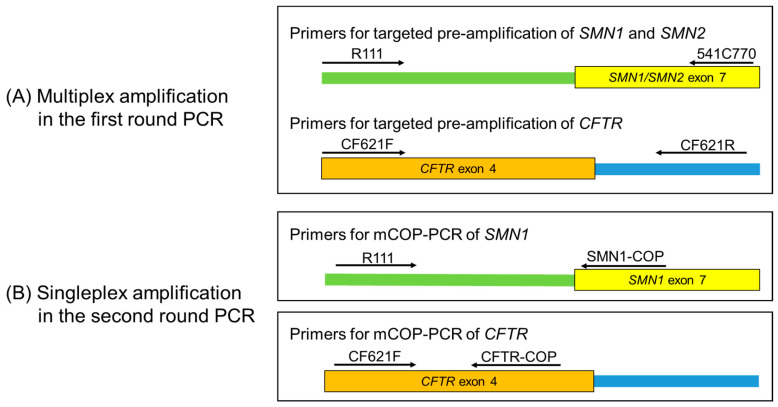
Primer positions used in the quantitative nested PCR. (**A**) Multiplex PCR in the first round of the nested PCR. Long *SMN1*, *SMN2*, and *CFTR* fragments were co-amplified by outer primers in a multiplex PCR. (**B**) Singleplex PCR in the second round of the nested PCR. Shorter fragments of *SMN1* and *CFTR* were separately amplified in different tubes. Primer positions and directions are indicated by arrows.

**Figure 2 IJNS-06-00043-f002:**
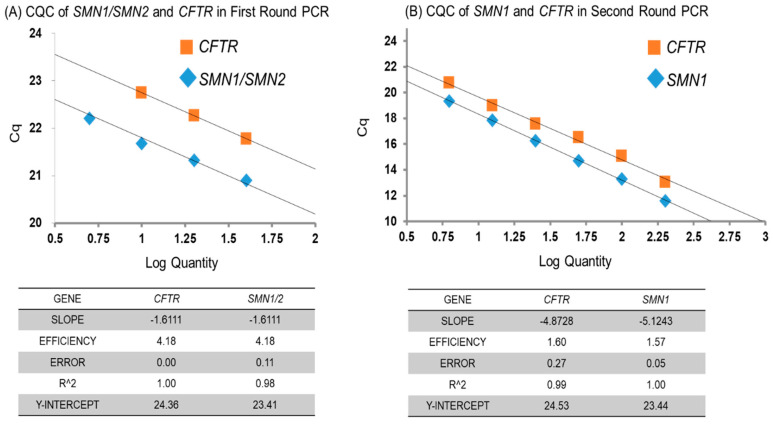
Cq-Log quantity curves (CQC). (**A**) Cq-Log quantity curves of *SMN1/SMN2* (*SMN1* and *SMN2* fragments amplified by the same primers) and *CFTR* in the first round PCR. The amplification efficiencies of the *CFTR* and *SMN1/SMN2* were similar. (**B**) *SMN1* and *CFTR* in the second round PCR. The amplification efficiencies of the *SMN1* and *CFTR* were also similar.

**Figure 3 IJNS-06-00043-f003:**
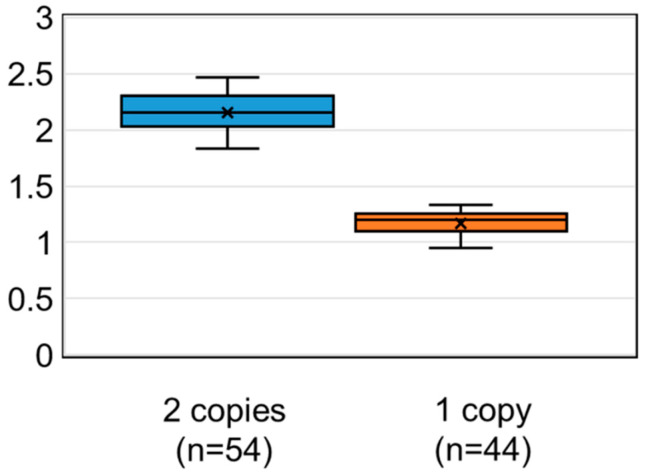
Calculated *SMN1* copy number values using dried blood spots (DBS) on Flinders Technology Associates (FTA) cards. Box-and-whisker plots of the calculated copy number values were obtained from 54 DBS samples with two *SMN1* copies and 44 DBS samples with one *SMN1* copy. The x mark indicates the average value of each group.

**Figure 4 IJNS-06-00043-f004:**
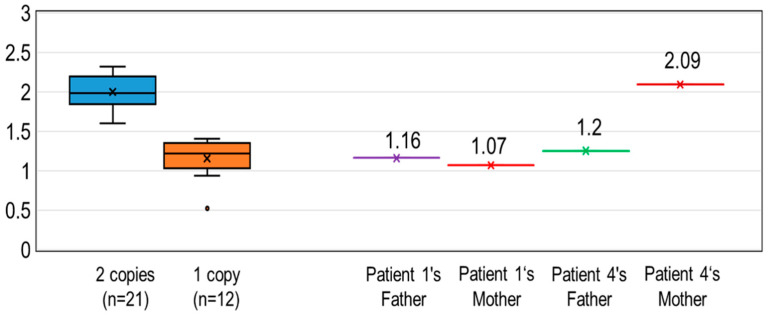
Calculated *SMN1* copy number values using DBS on Guthrie cards. Box-and-whisker plots of the calculated copy number values were obtained from 21 DBS samples with two *SMN1* copies and 12 DBS samples with one *SMN1* copy. Patient 1’s parents and Patient 2’s father carried one *SMN1* copy, but Patient 2’s mother carried two *SMN1* copies. The x mark indicates the average value of each group.

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
