# Peer review of "Assessment of Spinal Muscular Atrophy Carrier Status by Determining SMN1 Copy Number Using Dried Blood Spots"

_2409-515X, 2020, doi:10.3390/ijns6020043_

Round 1

Reviewer 1 Report

The manuscript of Wijaya et al report a method to detect bi-allelic SMN1 absence or one SMN1 copy (heterozygous state) by means of a previous published methodology in the same Journal (Int. J. Neonatal Screen. 2019, 5, 41; doi:10.3390/ijns5040041)

The manuscript is well written and illustrated. However, most of the methodology has been previously reported. The authors add and consider in this paper known modifications and calculations to include cases with one SMN1 copy.

My comments are as follow

  • Some repetitive phrases with the previous work are unavoidable given the similarity of some parts of the reports. Can the authors focus on the detection of heterozygotes and the limitation of the method? It is unclear the difference with the previous paper from the methodological point of view. This may strengthen the message of this paper.
  • There are clear limitations in detecting carriers by newborn screening. The specific application of this method (NS? Population carrier testing?) is not totally uncovered in this paper. Newborn screening and alternative methods of prevention such as population based carrier screening has been recently discussed in an International workshop (see PMID: 31882184)
  • No further discussion of cases with two SMN2 copies nor mention of possible alternatives to circumvent or minimize the residual risks are discussed (see PMID: 23788250 and PMID: 29904179)

Reviewer 2 Report

The objective of this article is stated as development of a system to detect heterozygous SMN1 deletion using dried blood spots and to confirm practicality of using Guthrie filter paper for resource limited situations. 

The first objective was accomplished.  However, one of the four families included as validation of the method had one parent with "2 + 0" SMN1 genes.  While this was discussed as a potential source of a false negative carrier screen, the presumed low incidence of this occurrence which would lessen the impact in population screening was not discussed. 

The second objective would require more detail than a simple statement that this is an "inexpensive method".  I am unaware of the relative cost of fluorescently-labeled oligonucleotide probes vs the cost of Eva Green Dye.  This should be discussed more expansively to make the point. 

Finally, the methodology here would appear to be similar to but technologically updated from the radioisotope method described in the referenced article by McAndrew etal (1997).  Perhaps this should be mentioned in section 4.4 where the authors create a distinction and identify their described methodology as a third approach to carrier detection.

Round 2

Reviewer 1 Report

Changes have been done and most are in general satisfactory. 

Please consider that this method have been published previously and is not novel. 

Authors should stress that this is a modified version of a previous method already published. 
